# Crosstalk of Hedgehog and mTORC1 Pathways

**DOI:** 10.3390/cells9102316

**Published:** 2020-10-18

**Authors:** Lasse Jonsgaard Larsen, Lisbeth Birk Møller

**Affiliations:** Department of Clinical Genetics, Copenhagen University Hospital, Rigshospitalet, Gl. Landevej 7, 2600 Glostrup, Denmark; Lasse.Jonsgaard.Larsen@Regionh.dk

**Keywords:** TSC, primary cilia, GLI1, GLI2, GLI3, S6K, 4E-BP1, eIF4E, autophagy, Mammalian target of rapamycin

## Abstract

Hedgehog (Hh) signaling and mTOR signaling, essential for embryonic development and cellular metabolism, are both coordinated by the primary cilium. Observations from cancer cells strongly indicate crosstalk between Hh and mTOR signaling. This hypothesis is supported by several studies: Evidence points to a TGFβ-mediated crosstalk; Increased PI3K/AKT/mTOR activity leads to increased Hh signaling through regulation of the GLI transcription factors; increased Hh signaling regulates mTORC1 activity positively by upregulating *NKX2.2*, leading to downregulation of negative mTOR regulators; GSK3 and AMPK are, as members of both signaling pathways, potentially important links between Hh and mTORC1 signaling; The kinase DYRK2 regulates Hh positively and mTORC1 signaling negatively. In contrast, both positive and negative regulation of Hh has been observed for DYRK1A and DYRK1B, which both regulate mTORC1 signaling positively. Based on crosstalk observed between cilia, Hh, and mTORC1, we suggest that the interaction between Hh and mTORC1 is more widespread than it appears from our current knowledge. Although many studies focusing on crosstalk have been carried out, contradictory observations appear and the interplay involving multiple partners is far from solved.

## 1. Introduction

Crosstalk between the Hedgehog (Hh) and Mammalian target of rapamycin complex 1 (mTORC1) signaling pathways has been hypothesized by the findings that both the Hh and mTORC1 signaling pathways are involved in the progression and metastasis of several cancers [1]. However, crosstalk between Hh and mTORC1 signaling has also been suggested in other systems. 

Canonical Hh signaling in vertebrates is one of the most studied signaling pathways connected to the primary cilium and it is well established that a functional primary cilium is required for Hh signaling [2,3]. Recent evidence also points to the primary cilium as a regulator of mTORC1 signaling. mTORC1 is regulated by TSC1 and TSC2, and a major part of the evidence for this connection has been obtained by investigating the effect of *Tsc1* and *Tsc2* knockdown (KD) in animal models. Thus, *Tsc1* KD in zebrafish leads to phenotype characteristics of ciliary mutants [4] and mouse embryonic fibroblasts (MEFs) lacking either *Tsc1* or *Tsc2* display a phenotype of enhanced ciliary formation [5]. Our own results also indicate a connection between mTORC1 and the primary cilium, as we observed an elongated ciliary phenotype in cells lacking *Tsc1* and a shortened ciliary phenotype in cells lacking *Tsc2* [6]. 

In the following sections we will shortly introduce the primary cilium and the Hh- and mTORC1 signaling pathways, followed by a summary of suggested mechanisms leading to observed Hh/mTORC1 crosstalk. 

## 2. Background

### 2.1. The Primary Cilium

The primary cilium is an immobile, microtubule-based organelle that projects as an antenna-like structure from virtually all adherent cells in the mammalian body during cellular quiescence. It functions as a sensory organelle and as a signaling compartment, by local regulation of receptors, including the G-protein coupled transmembrane receptors (GPCRs) [2,7,8]. The primary cilium coordinates a variety of signal transductions, including canonical Hh, mTOR, Wingless/Int, Platelet-Derived Growth Factor Receptor α, and Transforming growth factor β (TGF-β) signaling pathways, which are critical for the regulation of polarity, differentiation, migration, and proliferation of cells during embryonic development and in tissue homeostasis. The cilium is composed of a basal body (BB) and a microtubule cytoskeleton called the axoneme, surrounded by a lipid bilayer membrane with a strictly regulated receptor composition (Figure 1) [7,8].

The BB grows from the older of the two centrioles in quiescent cells (G0 phase). The two centrioles are present in the centrosome, which is the primary microtubule organizing center (MTOC) [9]. MTOC coordinates cell proliferation by facilitating the organization of the bipolar spindle poles during mitosis. The centriole pair is duplicated and segregated during each cell cycle, and as they are required for both ciliogenesis and cell cycle progression, primary cilia only assemble when cells exit the cell cycle. Centriole duplication is a tightly regulated process that normally occurs only once per cell cycle.

The mechanisms underlying the trafficking of receptors and signaling molecules to and within the cilium are not well understood. The movement is partly facilitated by intraflagellar transport (IFT) [2,3,10,11,12,13]. IFT, which is critical for the assembly and maintenance of cilia, involves motor-driven transport of large complexes called IFT-particles along the axoneme. These particles are composed of at least 30 proteins organized in IFT-A, IFT-B, and BBSome subcomplexes [13]. The IFT-B complex is linked to a kinesin-II motor, KIF3, for anterograde transport towards the ciliary tip, whereas the IFT-A complex is connected to a dynein motor, Dync2h1, for retrograde transport towards the ciliary basis.

### 2.2. The Hh Signaling Pathway

The Hh signaling pathway is fundamental for embryonic development and for postnatal tissue homeostasis; renewal and regeneration [14,15,16]. Dysregulated Hh pathway activity gives rise to birth defects, including left–right asymmetry of vertebrate embryos, and Hh signaling is associated with tumorigenesis. The Hh transduction pathway is very complex. Different, not well-defined modes of Hh signaling exist, which most commonly are classified as either non-canonical or canonical. 

Non-canonical Hh often refers to signaling that does not lead to activation of Glioma-associated oncogene (GLI) transcription factors [17]. Non-canonical Hh signaling pathways has been divided into two groups: *Type I* signals, Smoothened (SMO)-independent signals that stem from the receptor Patched1 (PTCH) and *Type II* signals, that are SMO-dependent. SMO-independent activation of GLI has also been classified as non-canonical Hh signaling. Hh signaling in vertebrates, involving Hh-dependent activation of GLI transcription factors via the SMO-to-GLI route, is named canonical Hh [18,19].

Vertebrate canonical Hh signaling is initiated by binding of a proteolytically processed and lipid-modified Hh ligand to its receptor, PTCH. Three different Hh-related genes are present in vertebrates, *sonic-hedgehog (SHh),*
*desert-hedgehog*, and *indian-hedghog* [20]. Hh ligand is palmitoylated and modified by cholesterol binding [21]. In the presence of the Hh ligand, PTCH is displaced away from the primary cilium, which allows accumulation of SMO in the ciliary membrane. The presence of activated SMO in the cilia leads to ciliary export of GRP161. 

The GPCRs, SMO and GPR161, regulate the activity of the GLI transcription factors, in opposite directions, by regulating the level of the second messenger cAMP in the primary cilium [22]. In the absence of Hh ligands, GPR161 catalyzes the GDP–GTP exchange of Gαs, leading to activation of adenyl cyclase (AC) and high levels of cAMP [23]. The presence of cAMP activates protein kinase A (PKA), which phosphorylates GLI2 and GLI3 (Figure 1). PKA phosphorylation (priming phosphorylation) triggers a cascade of GLIø3 phosphorylation by Glycogen Synthase Kinase 3-β (GSK3β) and casein kinase 1 (CK1) at multiple sites [24,25,26,27]. GLI2 and GLI3 each contain six PKA phosphorylation sites in their carboxy (C) termini (sites P1–P6). PKA phosphorylation of sites P1–P4 are required for GLI2 and GLI3 C-terminal processing to generate repressor forms (GLI2Rep and GLI3Rep), whereas phosphorylation of the P5–P6 sites inhibits transcriptional activity of GLI2 and GLI3. GLI1 also contains PKA phosphorylation sites (P1–2). Phosphorylation of GLI1 is required for restricting GLI1 activity [28,29]. Phosphorylation by PKA, GSK3β, and CK1 generates binding sites for ligases. GLI ubiquitylation promoted by E3 ligases leads to either proteasome-dependent proteolytic cleavage of the GLI2 and GLI3 factors [25,26] or degradation of GLI1 [30]. The formation of GLI2Rep is relatively limited compared to the formation of GLI3Rep, which explains why GLI2 is the primary mediator of Hh signalling, whereas GLI3 largely serves as a repressor [31]. The differences in the degree of processing have been shown to be due to sequence differences between the processing determinant domains located in the C terminal sequences of the GLI proteins [32]. Agonist activated SMO catalyzes the GDP–GTP exchange of all members of the G inhibitory (Gi) (Gαi1, Gαi2, Gαi3, Gαo, and Gαz), leading to inhibition of AC and reduced concentration of cAMP, which permits stabilization of the full length (FL) transcription factors GLI2FL and GLI3FL. The factors GLI2-FL and GLI3-FL are subsequently, by phosphorylation, activated (GLI2-A and GLI3-A) and translocated to the nucleus, and transcriptional activation of the Hh target genes including *GLI1* and *PTCH* are initiated [33,34,35]. Expression of *GLI1* is routinely used as a marker for Hh activity [6].

Chemical agonists that directly bind to SMO, such as SMO agonist (SAG) or purmorphamine also lead, independently of Hh and PTCH, to SMO accumulation in the cilium [36]. Antagonists such as cyclopamine, or blocking ubiquitination of SMO by an E1 ligase inhibitor, also provoke SMO accumulation in the cilium, but in this case without triggering downstream signaling as the SMO is inactive [37].

In the absence of Hh signaling, suppressor of fused (SUFU) has been demonstrated to regulate GLI negatively, whereas Hh signaling relieves this inhibition, but the mechanisms are far from solved. It is possible that SUFU primarily regulates GLI negatively by sequestering GLI in the cytoplasm, and in this way hampers nuclear localization [38,39]. However, it is also possible that SUFU prevents the transcriptional activity of GLI1 in the nucleus [40], but the exact role of SUFU in the nucleus is also under debate. Some studies suggest that SUFU dissociates from GLI in the presence of Hh stimulation, allowing transcriptional activity of GLI [41], whereas others suggest that GLI and SUFU remain associated and SUFU acts as a positive regulator of GLI1 and is essential for stabilization of GLI2/3FL [42] (Figure 1). The negative impact of SUFU on GLI activity is relieved by Hh signaling involving phosphorylation by PKA and GSK3β [43,44].

### 2.3. The mTOR Signaling Pathway

The mTORC1 complex is part of the PI3K/AKT/mTOR pathway [5,45]. mTOR activation is observed in many types of cancers [46,47,48,49]. The mTOR pathway is a central regulator of mammalian metabolism and physiology. It belongs to the phosphoinositide 3-kinase (P13K)-related kinase family. mTOR regulates cell growth, protein synthesis, proliferation, and autophagy by integrating signals from growth factors, hormones, nutrients, and cellular energy levels. mTOR is a member of the mTORC1 and the mTORC2 protein complexes, each composed of mTOR, Deptor, and mLST8 subunits. The mTORC1 complex contains in addition RAPTOR and PRAS40, while the mTORC2 complex contains three additional components, RICTOR, mSIN1, and PROTOR1 [50] (Figure 2).

Loss of either *TSC1* or *TSC2* causes constitutive mTOR activation and development of hypomelanotic macules [51]. The two proteins, hamartin encoded by *TSC1* and tuberin encoded by *TSC2* (here named TSC1 and TSC2, respectively), form a heterodimeric complex that inactivates mTORC1 via the small G-protein Ras homologue enriched in brain (RHEB) [52]. The mTORC1 inhibitor rapamycin and derivative compounds (rapalogues) are used for treatment of various types of cancer, in addition to Tuberous Sclerosis Complex (TSC), an autosomal dominant disorder caused by pathogenic variants in either *TSC1* or *TSC2* [53]. The mutated oncogenes, e.g., P13K and Ras and tumor suppressors, e.g., PTEN, LKB1, and NF1, are all members of the mTORC1 signaling pathway [54,55]. Activation of mTORC1 begins when, e.g., growth factors activate PI3K. PI3K activates AKT1 [56], which inactivates the TSC1/2 complex. When released from TSC inhibition, RHEB activates mTORC1 [57,58], which regulates the activity of 70 KDa ribosomal protein S6 kinase (S6K) and eukaryotic translation initiation factor 4E binding protein 1 (4E-BP1), leading leading to mRNA translation, cell growth, and cell proliferation [59]. Phosphorylation of 4E-BP1 inhibits its binding to eukaryotic translation initiation factor 4E (eIF4E). The factor eIF4E is involved in recruitment of the ribosomes to the cap structure of the mRNAs. Non-phosphorylated 4E-BP1 interacts strongly with eIF4E, thereby preventing translation. 

mTORC1 is located at the lysosomes, to RAB7-positive vesicles, and this location is required for its activation [60]. The lysosomal localization of active mTORC1 is dependent on Rag GTPases and the protein complex Ragulator [61,62]. The Rag GTPases form heterodimers (A or B together with C or D) and the active complex consists of RagA/B-GTP with RagC/D-GDP. This complex binds Raptor, leading to localization of mTORC1 to the lysosomal membrane. Activation of Rag GTPases and mTORC1 are primarily regulated by the presence of amino acid and glucose [63]. The regulator RHEB is located at the endomembrane system, including the lysosomes [64,65] (Figure 2). 

The mTORC1 signaling cascade receives inputs from various upstream cues related to the metabolic and nutritional status of the cell and translates these into regulation of cell growth. The mTORC1 complex receives input according to the energy status via AMP-activated protein kinase (AMPK), mitochondria, and lysosomes [64]. In the presence of nutrients, mTORC1 is activated and promotes cell growth, including protein synthesis, and energy storage. In contrast, during starvation, mTORC1 is inhibited and autophagy is induced by AMPK, leading to generation of intracellular nutrients and energy during degradation of non-functional or non-essential organelles or protein aggregates, contributing to cell survival [64]. Furthermore, inhibition of mTORC1 might also lead to increased apoptosis. Apoptosis is a regulated cell death. Reduced caspase activation and elevated protein expression of X-linked inhibitor of apoptosis proteins (XIAPs) have been associated with activated AKT in many cancers [66,67] (Figure 2).

The AMPK complex is sensitive to the cellular AMP/ATP and ADP/ATP ratios. AMPK is a protein complex consisting of α, β, and γ subunits. When the AMP or ADP concentrations are high, AMPK is activated by phosphorylation of protein kinases, e.g., LKB1 [68]. The activated kinase AMPK phosphorylates GLI1, leading to degradation of GLI1 and reduced Hh activity [69] (Figure 1). In addition, AMPK has been shown to inhibit mTORC1 signaling by enhancing the activity of TSC2 by activating phosphorylation [70]. This phosphorylation of TSC2 by AMPK primes the subsequent phosphorylation of TSC2 by GSK3β [71]. AMPK also inhibits mTORC1 activity by inhibitory phosphorylation of Raptor, which permits 14-3-3 binding to Raptor [72]. 

Autophagy is upregulated by AMPK-mediated phosphorylation of ULK1, leading to increased activity. The activated ULK1 upregulates the subsequent autophagy by phosphorylation of Raptor, leading to inactivation of mTORC1 [64,73]. Downregulation of autophagy, on the other hand, is regulated by mTORC1-mediated inhibitory phosphorylation of the ULK1-mAtg13-FIP200 complex [64,73,74] (Figure 2). 

## 3. Mechanisms Leading to Hh/mTOR Crosstalk

### 3.1. Hh and mTOR Crosstalk in Cancer

*GLI* has been found amplified and expressed in several tumor types, including rhabdomyosarcoma [75,76,77,78,79], and treatment of a human rhabdomyosarcoma cell line with the mTOR inhibitor rapamycin inhibits its growth. In similarity, it has been reported that SHh can increase epithelial to mesenchymal transition and metastasis through the PI3K/AKT/mTOR pathway [80,81,82]. The epithelial to mesenchymal transition was diminished by rapamycin in a xenograft model of rhabdomyosarcoma in a manner that inhibited both mTORC1 activity and Hh signaling, demonstrated by inhibition of growth and decreased expression of the Hh key genes: *GLI1*, *GLI2*, and *PTCH*, respectively [81]. 

In 2010, it was found that medulloblastoma resistance towards SMO inhibitors decreased as a result of combination therapy with SMO and PI3K/AKT inhibitors [83], and recently, the combination of the SMO inhibitor, NVP-LDE-225 with the PI3K/mTOR inhibitor, NPV-BEZ-235 exhibited improved efficacy towards inhibition of self-renewal and tumorigenicity of human pancreatic cancer stem cells, compared to treatment with only one of the agents, [84]. Furthermore, inhibition of PI3K/mTOR by NPV-BEZ-235 was shown to inhibit GLI1-dependent proliferation of androgen-independent prostate cancer cells [85]. Whereas the majority of prostate cancers depends on androgens (testosterone) during the initial stage, the cancer cells progressively lose the dependency on androgens during the advanced stages; thus, the cancer cells become resistant to hormonal treatments [86]. Interestingly, the authors found that the protein levels of GLI1 and GLI2 were highly increased in the androgen-independent prostate cancer cell lines PC3 and DU145 but not in the androgen-dependent cancer cell line LNCaP [85]. 

In summary, it is convincing that crosstalk between Hh and mTOR signaling in cancer cells exist, but the mechanisms are far from clear.

### 3.2. Hh/mTOR Crosstalk via TGF-β Signaling Pathway

We have recently demonstrated that mutations in either *Tsc1* or *Tsc2* inhibit Hh signaling [6]. We demonstrated that reduced Hh signaling in *Tsc1*-defective MEFs is due to a reduced amount of the Hh initiation transcription factor Gli2. Furthermore, the reduced amount of Gli2 was demonstrated to be a result of the reduced activation of the *Gli2* transcription factor, Smad2/3, and the reduced Smad2/3 activity, a result of a lack of functional Tsc1 required for TGFβ-induced activation of Smad2/3 [6]. However, a completely different mechanism appeared to cause reduced Hh signaling in *Tsc2*-defective cells. In contrast to *Tsc1*-defective cells, the reduced Hh signaling in *Tsc2*-defective cells could be corrected by treatment with the mTORC1 inhibitor, rapamycin. However, the molecular background for this is unknown. 

These results indicate that crosstalk between Hh and mTOR can be mediated via the TGF-beta pathway (Figure 3).

### 3.3. Influence of PI3K/AKT/mTOR on GLI Activity

Several studies have reported importance of the PI3K/AKT/mTOR pathway in regulating the cellular distribution of the GLI proteins. In 2006, a PI3K-dependent Akt activation was found to be essential for Hh signaling in the specification of neural fates in chick neural explants, chondrogenic differentiation of 10T1_2 cells, and GLI activation in NIH3T3 cells [87]. GLI2 phosphorylation and proteasomal degradation via PKA/CK1/GSK3β was shown to be inhibited by Akt activity, thereby promoting increased nuclear translocation of GLI2. The authors suggested various mechanisms by which Akt could regulate GLI2. One model was that Akt regulates a protein responsible for retaining GLI2 in the cytoplasm and thus Akt-mediated inhibition of this protein would result in nuclear translocation of GLI2 and allow GLI2 to escape PKA-mediated phosphorylation and subsequent degradation. Another possible explanation was that Akt targets proteins involved in the regulation of PKA activity and/or availability towards GLI2. It was reported to be an mTOR-independent mechanism, as rapamycin did not affect GLI2 activity [87].

A more recent paper showed that GLI1 and GLI2 were activated by Insulin like growth factor (IGF-1)-mediated activation of the PI3K/AKT pathway in renal cell carcinoma. IGF-1 induced both GLI1/2 mRNA and GLI-Luc reporter activity, in a non-canonical manner, that could be blocked by PI3K and GLI inhibitors, but not by the SMO inhibitor cyclopamine. Moreover, the combination of GLI and AKT inhibitors significantly suppressed renal cell carcinoma growth. However, the authors did not investigate whether these observations were mTOR-dependent [88]. 

Another study provided evidence that RAS/MEK and AKT signaling regulate the nuclear localization and transcriptional activity of GLI1 in melanoma and other cancer cells in response to SHh ligand. In melanoma, both the RAS–RAF–MEK–ERK (MAPK) pathway and the PI3K–AKT pathway are shown to be constitutively active [89]. Here, oncogenic RAS variants, and constitutively active MEK1 or AKT1, caused nuclear localization of myc-tagged GLI1 in the human melanoma cell line SK-Mel2. Further, blocking MEK or AKT activity increased cytoplasmic localization of GLI1 and reversed the effect of oncogenic RAS [90]. 

In the same direction, Glaucocalyxin A, which has been documented to induce apoptosis and suppress cell proliferation in various cancers, was recently demonstrated to exhibit anticancer effect on human osteosarcoma through a pro-apoptotic mechanism. Glaucocalyxin A was shown to increase apoptosis in human osteosarcoma cells, and this effect was accompanied by a concentration-dependent decrease in the nuclear concentration of GLI1, while the cytoplasmic concentration of GLI1 was increased. Glaucocalyxin A treatment also led to reduced expression of P13K and phosphorylated AKT, indicating reduced PI3K/AKT signaling. Notably, Glaucocalyxin A induced an inhibitory effect on the PI3K/AKT signaling pathway, as the decreased GLI1 nuclear localization and apoptosis were reversed by the P13K/AKT activator, IGF-1. Based on these results, it was concluded that Glaucocalyxin A inhibits GLI1 nuclear translocation via inhibition of the P13K/AKT pathway. The authors did not investigate a possible mTOR-dependent impact of this mechanism [91]. 

However, in contrast to *AKT1*, *AKT2* overexpression was reported to regulate GLI1 nuclear accumulation and transcriptional activity in the neuroblastoma cell-line, BE(2)-M17 negatively [92]. Whether this AKT2-dependent effect was mTOR-dependent was also not tested.

While the effect of mTOR was unknown or shown not to be responsible in the studies described above, mTOR-dependent regulation of GLI, through both S6K and 4E-BP1, has been demonstrated. In 2012, it was demonstrated that S6K1 phosphorylated GLI1 at p.Ser84. Through a co-immunoprecipitation experiment, the authors found that the interaction between GLI1 and SUFU was reduced in cells with the phosphor-mimetic p.Ser84Glu mutation of GLI1, and it was suggested that the GLI1 p.Ser84 phosphorylation could reduce GLI1 binding to SUFU and in this way be responsible for nuclear translocation of GLI1. The GLI1 phosphorylation at p.Ser84 was observed upon TNF-α treatment and was suggested to increase GLI1 oncogenic function in the development of esophageal adenocarcinoma, since cells with the GLI1 mutation, p.Ser84Glu had a higher level of colony formation in a soft-agar assay. Additionally, the described regulation of GLI1 activity was SMO-independent [93].

In contrast, a later study demonstrated that the impact of S6K1 on neuroblastoma cells was not mediated through GLI1, as KD of *S6K1* did not decrease the protein level of GLI1 and overexpression of *GLI1* could not rescue a reduced proliferation observed upon *S6K1* KD [94]. This indicates that cell type-specific mTOR signaling could be crucial for disease progression.

4E-BP1-mediated regulation of GLI1 has recently been demonstrated in cerebellar and medulloblastoma growth in mice. It was found that canonical Hh signaling required mTORC1-mediated inhibition of 4E-BP1, since the mTOR inhibitor Torin1 suppressed *GLI1* mRNA induction in response to SAG, and KD of *4E-BP1*, by shRNAs, reversed the effect of Torin1 on *GLI1* mRNA expression. Inhibition of mTORC1 further suppressed Hh-dependent growth of cerebellar and medulloblastoma [95]. It is interesting to note that Torin1 but not rapamycin could suppress SAG-mediated Hh signaling.

Skeletal muscles have high metabolic activity and reperfusion injury after e.g., ischemia or the use of a surgical tourniquet, to obtain a blood-free area, and may therefore cause damage and necrosis of skeletal muscles. The PI3K/AKT pathway is known to have a protective role against limb ischemia–reperfusion (I/R) injury [96,97,98,99]. Tourniquet-induced skeletal muscle I/R injury has been reported to activate AKT and mTOR through IGF-1 [100] and to induce Hh signaling through activation of the AKT/mTOR/S6K1 pathway in mice [101]. Here, the SHh protein level increased during reperfusion after 3 h of hindlimb ischemia. The SHh level peaked after five days of reperfusion. Furthermore, this increase was accompanied by an increase in Gli1 and Gli2 protein levels. The authors reported that the skeletal muscle fibrosis was aggravated by intraperitoneal injection of the SMO inhibitor cyclopamine, whereas it was decreased by overexpression of N-terminal human *SHh*. Inhibition of PI3K/mTOR by NVP-BEZ235 abrogated the SHh-induced skeletal muscle protection, indicating a protective role of mTOR-dependent Hh signaling in I/R injury [101].

In contrast, in another recent study, performed with overexpression of the *GLI* genes, the authors found that nuclear localization and transcriptional activity of GLI3, but not of GLI1 nor GLI2, in Human melanoma (SKMel29, MeWo), glioblastoma-astrocytoma (U373MG), and cervix carcinoma (HeLa) cells were regulated by rapamycin and protein phosphatase 2A (PP2A). PP2A is a protein phosphatase that acts in opposition to mTORC1 on the regulation of S6K and 4E-BP1. Inhibition of mTORC1 results in increased PP2A activity. An increase in PP2A activity led to cytosolic retention of GLI3 and reduced transcription of the GLI3 target gene, *Cyclin D1,* whereas reduced PP2A activity led to increased GLI3 nuclear localization and transcription of *Cyclin D1*. This cytosolic retention of GLI3 by rapamycin was dependent on a rapamycin-mediated activation of PP2A [102].

Together, these reports demonstrate that PI3K/AKT/mTOR can positively regulate Hh signaling, by increasing the activity of GLI1 and GLI2; also, the activity of GLI3 may be increased in case of decreased PP2A activity (and increased mTORC1 activity) (Figure 4). The involvement of mTOR was not investigated in all studies and the experimental procedures to investigate mTOR-dependence varied. Treatment with rapamycin was often the only assessment of mTORC1 dependence, even though rapamycin-resistant mechanisms of mTORC1 exist [103]. Thus, further studies are needed to reveal the underlying mechanisms through which mTORC1 can regulate GLI proteins in both canonical and non-canonical Hh signaling. 

### 3.4. Influence of Hh on PI3K/AKT/mTOR Activity 

A recent study reported a novel mechanism for crosstalk between SHh and PI3K/AKT/mTOR. The study identified Hh pathway mutations and structural copy number variants in patients with somatic overgrowth, macrocephaly, dysmorphic facial features, and developmental delay. These phenotypes resemble those of patients with PTEN mutations, suggesting that crosstalk between Hh signaling and mTORC1 could be involved in pathogenesis of the disease. Hh activating mutations of *PTCH* and *SUFU* resulted in increased PI3K signaling and increased phosphorylation of mTORC1 and 4E-BP1. Interestingly, decreased expression of several negative regulators of mTORC1 (*PRKAG2*, *RHOA*, *STRADB*, and *RRAGB*), leading to activation of mTORC1 was observed in human neural stem cells treated with SHh. When activated, GLI translocates to the nucleus and activates the expression of *LYF* and *NKX2.2*. The 5-kb upstream region of the negative regulators of mTOR (except *STRADB)* all contain transcription factor binding sites for LYF1-1 and NKX2.2. Hence, the authors suggested a model where increased NKX2.2 activity, mediated by GLI, decreases the expression of negative regulators of mTORC1 [104].

In addition to this, it has previously been demonstrated that Hh signaling modulates the expression of *AKT* genes, where *AKT1* was reported to be a direct transcriptional target of GLI1, with two putative binding sites for GLI1 in the *AKT1* promoter region [105].

Together, these results demonstrate that canonical Hh signaling can regulate mTORC1 activity positively, by downregulation of negative and upregulation of positive mTORC1 regulators. 

### 3.5. GSK3 Regulates Hh and mTOR Signaling

In mammals, GSK3 refers to the two paralogous proteins GSK3α and GSK3β, which are constitutively active serine/threonine protein kinases, involved in various cellular signaling, including Hh and mTOR signaling. The two kinases have specific substrates thus they cannot compensate for the loss of each other. However, due to the similarity of the two proteins, several chemical inhibitors of GSK3 target both GSK3α and GSK3β; thus, several studies do not discriminate between the two paralogs. GSK3β is inhibited via phosphorylation at p.Ser21 and activated via phosphorylation at p.Tyr279 [106,107]. The regulation of GSK3 depends also on the intracellular localization, cytoplasmic versus nuclear, as the localization affects the ability to phosphorylate target proteins. GSK3 preferentially targets proteins that have been phosphorylated by other kinases (i.e., after priming phosphorylation). 

As described above, GSK3β is involved in negative regulation of both Hh signaling and mTORC1 activity (Figure 1 and Figure 2). GSK3β has been shown to phosphorylate GLI transcription factors leading to their processing or degradation (Figure 1), and upon an AMPK-mediated priming phosphorylation of TSC2, GSK3β could phosphorylate TSC2, leading to activation, and thereby serve as a negative regulator of mTORC1/S6K activity (Figure 2) [71]. Consistently, it was later described that GSK3-mediated phosphorylation of TSC2, and thus inhibition of mTORC1 occurred during neural progenitor development [108]. 

Additional interactions of GSK3β with mTORC1 signaling, in different directions, have been reported. S6K2 and S6K1 were shown to inhibit GSK3β through p.Ser9 phosphorylation [106,109]. The effect of S6K1 was observed in TSC-deficient cells. This p.Ser9 located phosphorylation of GSK3β causes the N-terminal tail of GSK3β to act as a pseudo substrate that self-associates in the substrate binding pocket, thus hampering the binding and phosphorylation of substrates [107]. Importantly, this inhibition prevented GSK3β-mediated GLI1 degradation [109]. Furthermore, GSK3β regulates S6K1 activity positively through modulating phosphorylation of S6K1 at p.Ser371. Inhibition of GSK3β decreased the proliferation of rapamycin-sensitive breast cancer cells in an S6K1-dependent manner. However, it was also reported that GSK3β inhibition could decrease proliferation of rapamycin-insensitive breast cancer cells, suggesting that GSK3β can modulate cell proliferation by both S6K1-dependent and -independent mechanisms [110]. Additionally, GSK3β has been reported to directly phosphorylate, and thereby inactivate the mTORC1 substrate, 4E-BP1, leading to activation of eIF4E. Through this mechanism, GSK3β could bypass mTORC1 and thus enhance eIF4E-mediated protein synthesis and cell proliferation in a rapamycin-independent manner [111,112]. GSK3β was also suggested to stabilize mTORC1, through direct phosphorylation of Raptor at p.Ser859, since GSK3β inhibition or the p.Ser859Ala Raptor mutation reduced the interaction between the two mTORC1 components, mTOR and Raptor. This phosphorylation of Raptor at p.Ser859 might require a priming phosphorylation at p.Ser863 by an unknown kinase. Notably, the GSK3β-mediated phosphorylation of Raptor at p.Ser859 was amino acid-dependent, but the authors found no evidence of amino acid dependence on GSK3β activity, suggesting that the unknown priming kinase acts in an amino acid-sensitive manner [113]. 

Recently, active mTORC1 has been suggested to be responsible for the cytosolic localization of GSK3β, as mTORC1 inhibition was demonstrated to change the cellular distribution of GSK3β from the cytosol to the nucleus, leading to phosphorylation and degradation of the nuclear located GSK3β substrate, cMyc [114]. GSK3β phosphorylation of cMYC at p.Thr58, targets cMyc for ubiquitin-dependent proteasomal degradation [115]. This mTORC1-dependent control of GSK3β localization required normal late endosome/lysosome membrane trafficking, since a constitutively GDP-bound Rab7 mutant, which disrupts membrane trafficking at late endosome/lysosome, in similarity with inhibition of mTORC1, also led to loss of GSK3β from lysosomes and exhibited increased nuclear localization of GSK3β and a GSK3β-dependent reduced protein level of cMyc [114].

Since GSK3 regulates both Hh and mTORC1 signaling, GSK3 could be an important link of crosstalk between these pathways. However, this potential crosstalk could rely on the cellular localizations of GSK3 and the effectors of the Hh and mTOR pathways, since several components of Hh signaling localize to primary cilia and mTOR signaling is regulated at the lysosomes. In accordance with this, mTORC1 has the ability to regulate GSK3 localization (nuclear/cytosol distribution) [114]. These studies demonstrate that crosstalk between Hh, mTORC1, and GSK3 signaling could be significant for disease development (Figure 5). AMPK is also a member of both signaling pathways and therefore potentially an important link.

### 3.6. Hh and mTOR Crosstalk Mediated by DYRK

Several examples of crosstalk between the Hh and mTOR signaling pathways that emerged through the Dual-specificity and Tyrosine(Y)-regulated kinase (DYRK) family have been demonstrated. The DYRK protein family consists of five members: DYRK1A, DYRK1B, DYRK2, DYRK3, and DYRK4 [116,117,118,119,120,121]. The DYRK kinases phosphorylate several substrates that are involved in many cellular processes and they are thought to have essential functions during development and in maintaining homeostasis. These kinases are associated with several pathologies, including cancer [122]. DYRK1A was shown to promote GLI1 activity and nuclear translocation through direct phosphorylation of GLI1 [116,117,123,124]. However, DYRK1A was also reported to induce GLI1 degradation in a manner that was mediated by regulators of the actin cytoskeleton [117]. Furthermore, DYRK1A was shown to regulate mTORC1 activity negatively. Overexpression of *DYRK1A* increased phosphorylation and activity of both TSC1 and TSC2. In agreement with this, increased phosphorylation of S6K1 and 4E-BP1 was observed in *DYRK1A* KD cancer cells and this effect was inhibited by rapamycin [125].

Both negative and positive regulation of canonical Hh signaling have also been described for DYRK1B [118,119]. DYRK1B was demonstrated to work as a positive regulator in humane medulloblastoma cells (DAOY), as siRNA-mediated KD of *DYRK1B* eliminated SAG-induced GLI1 induction [119]. In contrast, another group demonstrated that SAG-treated NIH3T3 cells transfected with Dyrk1b expressed decreased Hh signaling, compared to the controls [118]. The same group also demonstrated, using the KD approach, a negative effect of Dyrk1b on Hh signaling, as the KD of *Dyrk1b* in a MEF cell line, stably expressing SHh ligand, resulted in upregulation of the Hh target genes with increased expression of *Gli1* mRNA [126]. Surprisingly, the authors also found that the basal level of the Gli1 protein, without Hh pathway activation, increased in NIH3T3 cells upon *Dyrk1b* overexpression. This increase in Gli1 protein was a result of increased Gli1 protein stability trough a mechanism that involved Akt activity, since inhibition of Akt reduced the level of Gli1 protein (non-canonical effect) [126]. An activation of Akt was observed in the *Dyrk1b* transfected cells [126]. 

The overexpression of *Dyrk1b* in NIH3T3 cells resulted in an increased mTORC1 pathway activation, verified by phosphorylation of mTor at p.Ser2448, of Akt at p.Ser473 and p.Thr308, of S6k at p.Thr389, and of S6 at p.Ser235/236. Since mTORC2 phosphorylates Akt at p.Ser473 and S6K/S6 are downstream targets of mTORC1 signaling, these results indicate a possible role of both mTORC1 and mTORC2 signaling in Dyrk1b-mediated regulation of Hh signaling [126].

In a recent study, *Dyrk2* knockout mice (*Dyrk2^−/−^*) were generated [127]. The homozygous *Dyrk2^−/−^* mice died at or close to birth, and multiple defects were observed, including defects in skeletal development. Investigation of the MEF cells revealed that Dyrk2 positively regulates Hh signaling as SAG-induced expression of *GLI1* was significantly decreased in *Dyrk2^−/−^*. The protein level of Gli2, and to some extent Gli3, was also reduced in SAG stimulated *Dyrk2^−/−^* MEFs, indicating that Dyrk2 is also important for the stability of the Gli2/3 proteins [127]. Dyrk2 has been reported to directly phosphorylate Gli2 and thus promote Gli2 degradation in NIH3T3 cells [120]. It is possible that Dyrk2 acts as a priming kinase and Dyrk2-mediated phosphorylation of Gli2 is followed by phosphorylation of Gli2 by CK1 and GSK-3β, leading to ubiquitin-mediated degradation of Gli2 [120].

Investigation of the Dyrk2^−/−^ MEF cells revealed that also the mTORC1 activity was increased [127]. Furthermore, an increase in the length of the primary cilia was observed in the *Dyrk2^−/−^* MEFs. Rapamycin did not affect the ciliary length [127]. DYRK2 has also, in another study, been demonstrated to repress mTORC1 activation as mTORC1 activation was increased, with increased phosphorylation of S6K and 4E-BP1, in *DYRK2* KD breast cancer cells. The effect of DYRK2 on mTORC1 was suggested to be through a mechanism of DYRK2-mediated phosphorylation of mTOR at p.Thr631, followed by degradation of mTOR through the ubiquitin proteasome pathway [128]. It was demonstrated that the mTOR inhibitor Everolimus, a rapamycin analog, could significantly inhibit tumor growth of DYRK2-depleted cells, compared to the control cells. Importantly, the sensitivity of Everolimus correlated with the expression of *DYRK2* in patients with hormone receptor-positive metastatic breast cancer. Patients with low expression of *DYRK2* (and high mTOR activity) had a higher clinical benefit than those with high expression of *DYRK2* [128].

Thus, whereas DYRK2 regulates Hh signaling positively, both positive and negative regulation of Hh has been observed for DYRK1A and DYRK1B. In contrast DYRK2 regulates mTORC1 negatively whereas both DYRK1A and DYRK1B regulate mTORC1 signaling positively.

### 3.7. Autophagy/Primary Cilium and Hh/mTOR Interaction

Serum starvation inhibits mTOR activity, and autophagy is induced [129]. Both the formation of primary cilia and mTOR signaling is regulated by nutrient availability generated by autophagy. Accordingly, primary cilia and Hh signaling have both been linked to autophagy, where reciprocal regulation has been observed between these cellular mechanisms. 

In 2013, autophagy was reported to regulate ciliogenesis. Upon serum starvation, the autophagic degradation of Oral–Facial–Digital Syndrome type 1 at the centriolar satellites, promoted primary ciliogenesis in MEFs [130]. Another study from 2013 reported that the basal level of autophagy had a negative role on ciliogenesis, through degradation, and thereby limiting the amount of IFT20 accessible for shuttling between the Golgi and the ciliary base. When autophagy was induced, as a result of a serum-depleted medium, IFT20 was not degraded. IFT20 is a component of IFT-B, which is crucial for ciliogenesis.

Contradictory observations have been published according to the interplay between Hh, autophagy, and mTORC1. A study discovered Hh to regulate autophagy in both mammals (HeLa and MEF cells) and *Drosophila*. They showed that Hh signaling inhibits autophagosome synthesis both in basal and autophagy-induced conditions (grown for 4 hours in HBSS with calcium and magnesium). This effect was dependent on Gli2, but insensitive to rapamycin [131]. In contrast, another study, performed in MEFs and mouse kidney epithelial cells, indicated that primary cilia, were necessary for serum starvation-induced autophagy. Also intact Hh was shown to be important for autophagy [132]. These observations were also insensitive to rapamycin and therefore concluded not to be dependent on mTORC1 signaling [132]. In 2015 it was demonstrated that both basal and starvation-induced autophagy was reduced in human kidney cells with ciliary deficiency (and probably also hampered Hh signaling) but, in this case, the authors concluded that the reduced autophagy was dependent on increased mTORC1 signaling, since rapamycin restored the autophagy [133]. 

Very recently, the IFT system was coupled with mTOR-dependent biogenesis of lysosomes in Jurkat cells. The authors reported that the IFT-B component IFT20 inhibits lysosomal biogenesis and function. Enhancement of lysosome biogenesis, observed in IFT20-deficient cells, was associated with upregulation of Transcription factor EB (TFEB) and the TFEB-dependent expression of the lysosomal gene network, in addition to reduced activity of mTOR and ERK1/2 [134]. TFEB-dependent transcription is essential for the regulation of lysosome biogenesis, and the nuclear localization of TFEB is negatively controlled by ERK and mTOR [135,136]. Interestingly, it was found that torin1-mediated inhibition of mTOR, but not inhibition of ERK, also led to increased TFEB-dependent gene transcription [134]. Thus, the ciliary trafficking protein, IFT20 was suggested to regulate lysosome biogenesis through a mTOR-dependent mechanism.

In addition, lysosomal mTORC1 is regulated by the Rag proteins [60,62]. It was recently found, through a yeast two-hybrid assay, that RagA interacts with WD repeat domain 35 (WDR35), a component of the IFT-A complex. This interaction was verified in 293T cells and indicates that WDR35 could be involved in regulation of both ciliary signaling and mTORC1 signaling. Notably, the authors found that *WDR35* overexpression decreased phosphorylation of S6 in a RagA-, RagB-, and RagC-dependent manner. However, the interaction between RagA and WDR35 seemed to be located at the ER and not at the lysosomes. Thus, this possible dual regulation might be mediated through subcellular translocation of proteins involved in both Hh and mTORC1 signaling, in a manner mediated by RagA and WDR35 interactions [137]. 

Together these studies demonstrate that an interaction between Hh signaling, autophagy, ciliogenesis, mTORC1, and lysosomal biogenesis exists. 

## 4. Conclusions

In summary, it is convincing that crosstalk between Hh and mTORC1 signaling exists. Observations from cancer cells strongly indicate crosstalk between Hh signaling and mTORC1. This assumption is supported by several studies. The mechanism behind the observed crosstalk might depend on the specific cell lines used. In Tsc1 defective cells the crosstalk between Hh and mTORC1 is mediated via the TGF-β pathway. Several reports demonstrated a mutual positive regulation between Hh and mTORC1. Increased PI3K/AKT/mTOR activity leads to increased Hh signaling, by increasing the activity of the Hh transcription factors GLI1/2/3. In similarity, increased Hh signaling positively regulates mTORC1 activity, by upregulation of *NKX2.2*, which is assumed to downregulate the negative regulators of mTOR (*PRKAG2, RHOA, STRADB,* and *RRAGB*), leading to activation of mTORC1. GSK3 and AMPK are members of both signaling pathways, and potentially important links. Both AMPK and GSK3 can inhibit Hh by phosphorylation of GLI1, promoting its degradation, and inhibit mTORC1 activity by activating phosphorylation of TSC2. The kinase DYRK2 positively regulates Hh signaling and negatively regulates mTORC1 signaling. In contrast, both positive and negative regulation of Hh signaling have been observed for DYRK1A and DYRK1B, which both regulate mTORC1 signaling positively. According to observed crosstalk between cilia, Hh and mTORC1, involving also regulation of cellular localization of the involved proteins, we suggest that the interaction between Hh and mTORC1 is more widespread than it appears from our current knowledge. Several contradictory observations have been published, indicating the complexity of this crosstalk. Different cell lines with different genetic mutations have been used, which might explain at least part of the divergence. Further research using more standardized systems is necessary.

## Figures and Tables

**Figure 1 cells-09-02316-f001:**
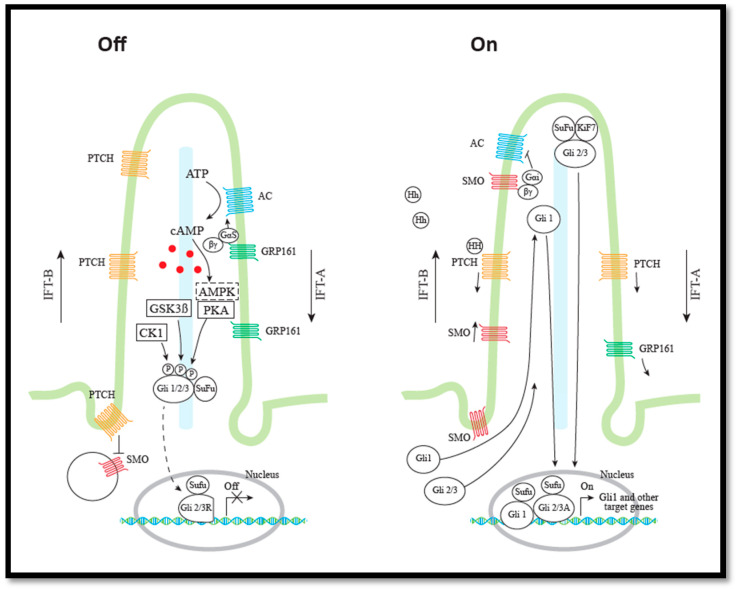
This is a model of Hh signaling in the primary cilium. To the left, Hh in the Off state. To the right, Hh in the On state. In the Off state, GRP161 and PTCH1 are in the primary cilia. GRP161 inhibits Hh signaling by activation of PKA, leading to phosphorylation of GLI. GLI2/3 are subsequently phosphorylated by CK1 and GSK3B and either proteolytically degraded to GLI2/3Rep or completely degraded. AMPK can also perform the priming phosphorylation of GLI. SUFU inhibits GLI. In the On state, Hh activation leads to removal of PTCH from the cilium, and entrance of SMO. The presence of active SMO in the cilium permits the export of GRP161. SMO prevents GLI degradation by downregulation of PKA. The full-length GLI1/2/3FL are accumulated and translocated to the nucleus in the active GLI1/2/3A form, leading to transcription of the Hh target genes (e.g., *GLI1, PTCH1, AKT, NKX2.2*) The IFT-B complex is involved in anterograde transport and the IFT-A in retrograde transport. See text for further details.

**Figure 2 cells-09-02316-f002:**
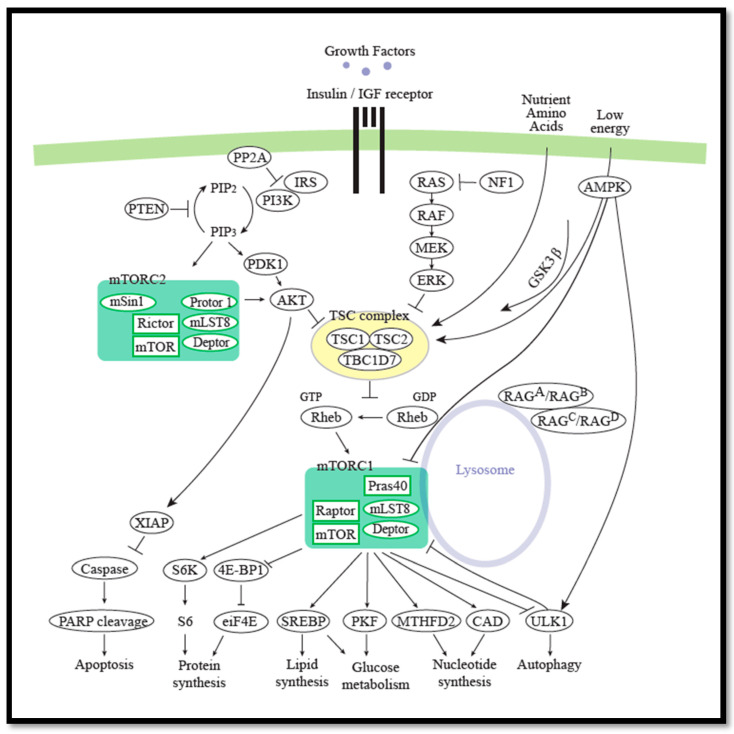
The mTOR pathway. mTOR is a member of the mTORC1 and the mTORC2 complexes. mTORC1 is composed of mTOR, Deptor, mLST8, RAPTOR, and PRAS40. mTORC2 is composed of mTOR, Deptor, mLST8, RICTOR, mSIN1, and PROTOR1. The tumor-suppressor TSC1/2 complex negatively regulates mTORC1 by inhibiting the protein RHEB. P13K, the RAS–RAF–MEK–ERK (MAPK) pathway, PTEN, LKB1, and NF1 are all members of the mTORC1 signaling pathway. Activation of mTORC1 begins when, e.g., growth factors activate PI3K. PI3K activates AKT1, which inactivates the TSC1/2 complex. RHEB activates mTORC1, which regulates the activity of S6K and 4E-BP1. The lysosomal localization of active mTORC1 is dependent on Rag GTPases. mTORC1 regulates nucleotide synthesis, glucose metabolism, lipid synthesis, and protein synthesis positively and regulates autophagy negatively. AMPK regulates mTORC1 negatively. Apoptosis is negatively regulated by AKT. See text for further details.

**Figure 3 cells-09-02316-f003:**
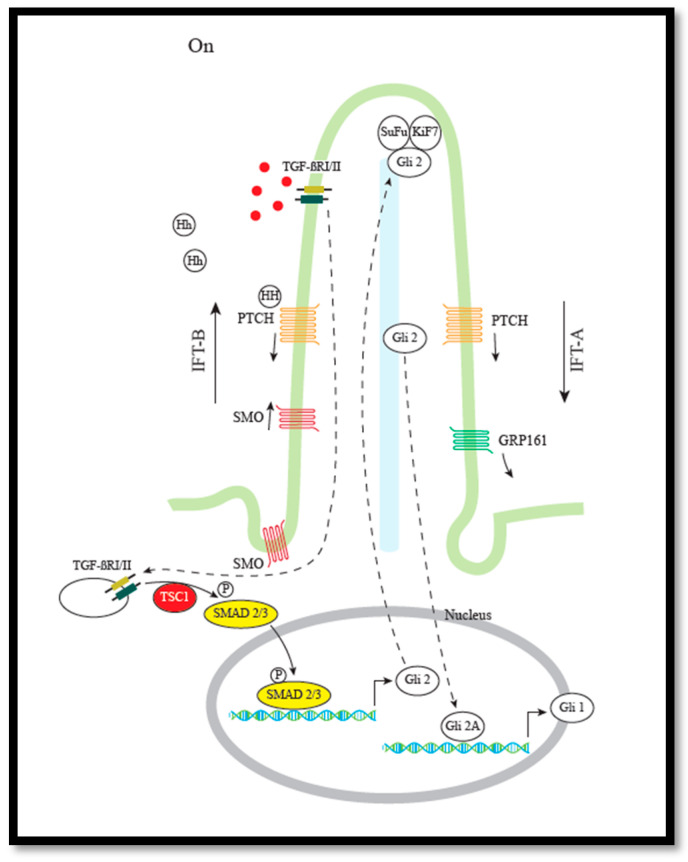
Hh and mTOR crosstalk via TGFβ signaling. The TGF-β-activated TGF-β receptor complex (TGF-βI/II) activates, by phosphorylation, the GLI2 transcription factor SMAD2/3. For this activation, TSC1 is required. See text for further details.

**Figure 4 cells-09-02316-f004:**
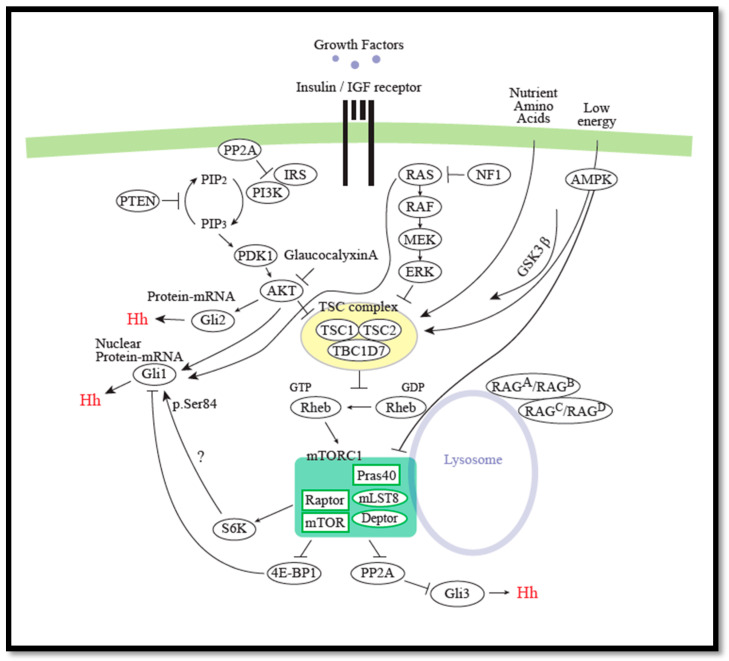
Connection between the PI3K/AKT/mTOR pathway and GLI activity. Increased Akt activity leads to increased GLI2 activity due to inhibition of degradation via PKA/CK1/GSK3β. IGF-induced activation of PI3K/AKT leads to increased expression of *GLI1/2*. Active RAS/MEK/AKT leads to increased nuclear localization of GLI1. Rapamycin inhibition of mTORC1 leads to increased PP2A activity, followed by decreased nuclear localization and transcriptional activity of GLI3 (but not GLI1 or GLI2). Glaucocalyxin A inhibits AKT. S6K1 phosphorylates GLI1 at p.Ser84, leading to increased nuclear location of GLI1. This effect could however not be confirmed by KD of *S6K1*. mTORC1-mediated inhibition of 4E-BP1 has been demonstrated to be required for canonical Hh signaling.

**Figure 5 cells-09-02316-f005:**
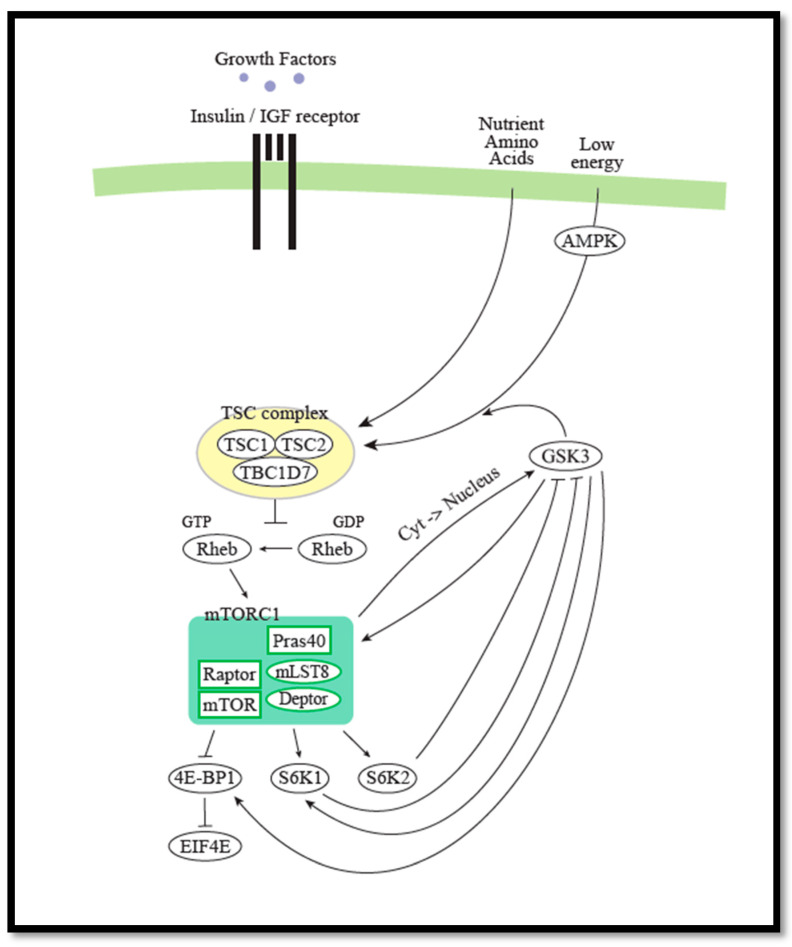
GSK3 in Hh and mTOR crosstalk. GSK3 inhibits Hh signaling by phosphorylation of GLI transcription factors, leading to their degradation. Upon an AMPK-mediated priming phosphorylation of TSC2, GSK3β could phosphorylate and activate TSC2, leading to inhibition of mTORC1. Inhibition of mTORC1 was demonstrated to change the cellular distribution of GSK3β from the cytosol to the nucleus. In contrast, GSK3β activates mTORC1, through direct phosphorylation and stabilization of Raptor. GSK3β also positively regulates S6K1 and negatively regulates 4E-BP1, by phosphorylation, in both cases leading to increased protein synthesis. S6K2 and S6K1 could both inhibit GSK3β through p.Ser9 phosphorylation.

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
