# Peer review of "Crosstalk of Hedgehog and mTORC1 Pathways"

_cells, 2020, doi:10.3390/cells9102316_

Round 1
Reviewer 1 Report
This is an interesting and well written review.
Requires minor modifications to the English.
Author Response
We have now tried to improve the english
Reviewer 2 Report
This review details interactions between Hedgehog signaling and mTORC1. The authors suggest that these are more widespread and critical than previously estimated. This point is well supported by many studies cited by the authors in their review.
My major concern is that the review tends to detail too many separate studies, without summing up the information in a manner that can provide the reader with a clearer picture of the subject. Although admittedly, crosstalk between different signaling pathways tend to be 'messy', as they often involve positive and negative loops, the review fails to provide the emphases that will make it easier for the reader to make order out of this mess. In this regard, it will help if the authors will (1) provide more illustrations for the new data that links the discussed signaling pathways/proteins. In this regard, figures 1 and 2 are not very helpful. They introduce Hh and mTORC1 signaling, as has previously done in many different reviews. Only figure 3 partly summarizes the interactions discussed in the review. Also, a conclusion paragraph is missing.
The abstract can be improved by using less specific terminology and by defining new signaling pathways/proteins mentioned.
Extensive editing of English language and style is required.
For example:
- Abstract – Crosswalk should be crosstalk
- Abstract- Hh should be defined when first introduced.
- Abstract – an incomplete sentence "according to DYRK a very complex pattern is formed".
- Line 30 "many years ago" – it is preferable that the authors give a better time frame.
- Line 51 – TSC1 is mentioned here for the first time without stating the link between TSC1 and the TOR pathway.
- Line 220 – a reference is missing. There are reviews on the subject that can be cited.
Author Response
(1) We have now added more illustrations
2) A conclusion is added after each chapter and at the end
3) The abstract is improved
4) Extensive editing of English language and style is performed .
Reviewer 3 Report
In the present review, the authors reported the current knowledges on the crosstalk between Hedgehog and mTORC1 signaling pathways. Although the topic discussed in this work is very interesting, the manuscript requires an extensive revision to be accepted for publication.
- My major concern is the drafting of the manuscript. It is poorly formatted and the grammar is confusing throughout. It is difficult to follow and the meaning of some sentences is not clear (i.e. line 16: According to DYRK a very complex pattern is formed”; line 30:” Many years ago, GLI has been found amplified and expressed in several tumor types including rhabdomyosarcoma and a human rhabdomyosarcoma cell line, were shown sensitive to the mTOR inhibitor, Rapamycin, as Rapamycin treatment inhibits the growth; line 545: Thus, an interaction between Hh signaling and autophagy, between ciliogenesis and autophagy, 546 between and ciliogenesis, autophagy and mTOR, between lysosomal biogenesis, between ciliogenesis 547 and mTOR and, between ciliogenesis and subcellular location”).
- The introduction should include a general description of the field of investigation on which the review is focused and not specifically reports experimental evidences supporting the crosstalk between Hedgehog and mTORC1. In this section, are mentioned pathways components not yet described in the manuscript.
- The sections regarding the description of the pathways are poorly structured. For example, for the Hh pathway, the authors first mentioned the cilium, then they introduced the repression forms of the transcription factors returning immediately afterwards to the description of the activation of the signaling.
- “GSK3 Mediated Hh and mTOR crosstalk” is just a guess of the authors. GSK3 is a common regulator of the two pathways rather than a mediator of their crosstalk, the title of the section should be modified.
- References should be uniformed and reported in the same way. Moreover, some names are not written properly or are not even reported (i.e. ref 28 and ref 76).
Author Response
- Thank you so much for all the comments
- We have performed corrections of the language. We hope it is much better now.
- We have changed the introduction. We belive it is easier to follow now
- We have changed the title of the GSK chapter
- We have added more figures and conclusions
- We have improved the references
- We have performed all the suggested corrections
Round 2
Reviewer 2 Report
The writers have done considerable work in trying to improve their review article. However, the article still details too many separate studies and lacks focus. In addition, there are still typos and English grammar mistakes.
Reviewer 3 Report
The authors addressed my comments by making the appropriate changes to the manuscript.